# Doxycycline-Loaded Nitric Oxide-Releasing Nanomatrix Gel in Replanted Rat Molar on Pulp Regeneration

**Kwan-Hee Yun** [1], **Mi-Ja Ko** [1], **Yong-Kown Chae** [1], **Koeun Lee** [2], **Ok-Hyung Nam** [3], **Hyo-Seol Lee** [3], **Kyounga Cheon** [4] and **Sung-Chul Choi** [3,*]

[1] Department of Pediatric Dentistry, Graduate School, Kyung Hee University, Seoul 02447, Korea; berryfield81@naver.com (K.-H.Y.); angelko2@naver.com (M.-J.K.); van_0418@naver.com (Y.-K.C.)

[2] Department of Pediatric Dentistry, Kyung Hee University Dental Hospital, Seoul 17104, Korea; olivedlr@naver.com

[3] Department of Pediatric Dentistry, School of Dentistry, Kyung Hee University, Seoul 02447, Korea; pedokhyung@khu.ac.kr (O.-H.N.); stberryfield@gmail.com (H.-S.L.)

[4] Department of Pediatric Dentistry, University of Alabama at Birmingham, Birmingham, AL 35294, USA; kcheon@uab.edu

\* Correspondence: pedochoi@khu.ac.kr; Tel.: +82-2-958-9339

**Abstract:** The aim of the present study was to evaluate the effect of doxycycline-loaded NO-releasing nanomatrix gel on pulp regeneration in replantation of avulsed rat teeth. A total of 28 maxillary first molars extracted from rats were replanted. The rats were divided into two groups based on the use of root surface treatment: doxycycline-loaded NO-releasing nanomatrix group and no treatment. Eight weeks after replantation, the rats were sacrificed, and the teeth were evaluated using histomorphometric analysis. On histomorphometric analysis, the NO-releasing nanomatrix group demonstrated a significantly lower grade of pulp inflammation ($1.00 \pm 1.11$, mean $\pm$ standard deviation) compared to the no treatment group ($2.21 \pm 1.25$, $p = 0.014$). NO-releasing nanomatrix group showed a significantly higher grade of pulp regeneration ($2.57 \pm 0.85$, $p = 0.012$) and significantly lower grade of pulp inflammation ($1.00 \pm 0.68$, $p = 0.025$) compared to the no treatment group. In conclusion, NO-releasing nanomatrix gel improved pulp regeneration of replanted teeth, though the sample size of this study was rather small. Within the limits of this study, NO-releasing nanomatrix gel can provide more favorable pulpal regeneration despite replantation.

**Keywords:** avulsion; replantation; nanomatrix gel; nitric oxide; pulp regeneration



## 1. Introduction

Tooth avulsion is a complete displacement of a tooth from its alveolar socket due to a traumatic dental injury (TDI) [1]. With tooth avulsion and replantation procedures, the damaged root surface structure and periodontium undergo inflammatory and infectious stages, often leading to a failure in functional recovery [2,3]. Therefore, it is recommended that the avulsed tooth should be replanted as soon as possible after the trauma and stabilized to allow healing [2,4].

For the replantation of an avulsed tooth, treatment goals are to promote healing without inflammation or infection and to promote the regeneration of the tooth's apex structure. However, there are no effective treatments to provide pulp regeneration with reduced inflammation [5–7]. In the literature, topical application of doxycycline to the root surface prior to replantation has been suggested and utilized to control microorganisms and minimize inflammatory responses [5,8]. In several animal studies, application of doxycycline prior to replantation has been reported to induce pulp regeneration, but complete regeneration of pulp tissue by doxycycline has not been achieved yet [5,6,9,10].

Recent advances in regenerative tissue engineering have enabled to recover the functional component of the original tissue as well as structural replacement of defects in oral

and craniofacial tissue [11–13]. This has been achieved by controlling the extracellular microenvironment, which plays a crucial role in tissue development. Regenerative tissue engineering approaches are based on the tissue engineering triad: cells, their extracellular matrix (ECM; scaffolds), and signals, which can be used individually or in combination to optimize regeneration and engineering of functional tissue [13].

Nitric oxide (NO) is a critical signaling molecule that modulates physiological responses in human body, including the inflammatory process. In acute inflammation, it regulates the progression of the inflammatory process and promotes angiogenesis during chronic inflammation [14–17]. Moreover, NO has excellent antibacterial and antiviral properties. Since NO has a short half-life, it requires a suitable delivery system for clinical use [17,18].

Recent literature demonstrated that NO possesses regenerative potential in cardiomyocytes and islet cells [19–21]. Furthermore, a previous study reported that NO-releasing nanomatrix gel promoted pulp revascularization potential in comparison with the conventional regenerative endodontic procedure [22]. NO stimulates angiogenesis in endothelial cells and can increase the survival of endothelial cells [23]. Additionally, endothelial cells in pulp tissue proliferate at the pulp injury site and stimulate angiogenesis with the interaction of fibroblasts [24]. Thus, it was hypothesized that addition of NO could promote pulp regeneration after replantation of avulsed rat teeth. The present study's results confirmed the hypothesis.

A biomimetic nanomatrix gel is formed by self-assembled peptide amphiphiles (PAs), which consist of a hydrophilic functional peptide sequence attached to a hydrophobic alkyl tail [25]. They can be easily synthesized and are chemically stable. In addition, they can form a viscoelastic gel by themselves in certain circumstances [25,26]. This gel can be used to trap and/or isolate cells, support their growth, and provide them with a controlled environment mimicking natural ECM. Several studies have reported the promising potentials of PAs as a biomaterial for use as a drug delivery system, including for NO molecules [19–21,27].

For these reasons, NO-releasing nanomatrix gel can be a promising candidate for replantation of an avulsed tooth. The aim of the present study was to evaluate the effect of doxycycline-loaded NO-releasing nanomatrix gel on pulp regeneration in the replantation of avulsed rat teeth.

## 2. Materials and Methods

### 2.1. Ethics Approval

The study was reviewed and approved by the Ethics in Institutional Animal Care and Use Committee of Kyung Hee Medical Center, Kyung Hee University, Seoul, Korea (KHMC-IACUC-16-022).

### 2.2. Synthesis of NO-Releasing Nanomatrix Gel

Synthesis of NO-releasing nanomatrix gel was performed as previously described [22]. Briefly, the gel was composed of a mixture of two types of PAs incorporated with NO gas: PA-YIGSR [CH3(CH2)14CONH-GTAGLIGQ-YIGSR] and PA-KKKKK [CH3(CH2)14CONH-GTAGLIGQ-KKKKK]. PA-YIGSR was composed of an endothelial cell adhesive ligand (YIGSR) coupled with a matrix metalloprotease-2 degradable sequence (GTAGLIGQ) to form PA-YIGSR. PA-KKKKK contained a NO donor polylysine (KKKKK) linked to the MMP-2 degradable sequence, forming PA-KKKKK. A mixture of PA-YIGSR and PA-KKKKK at a 9:1 molar ratio was reacted with NO gas to generate PA-YK-NO; then, gelation of the mixture followed (Figure 1) [21,28].

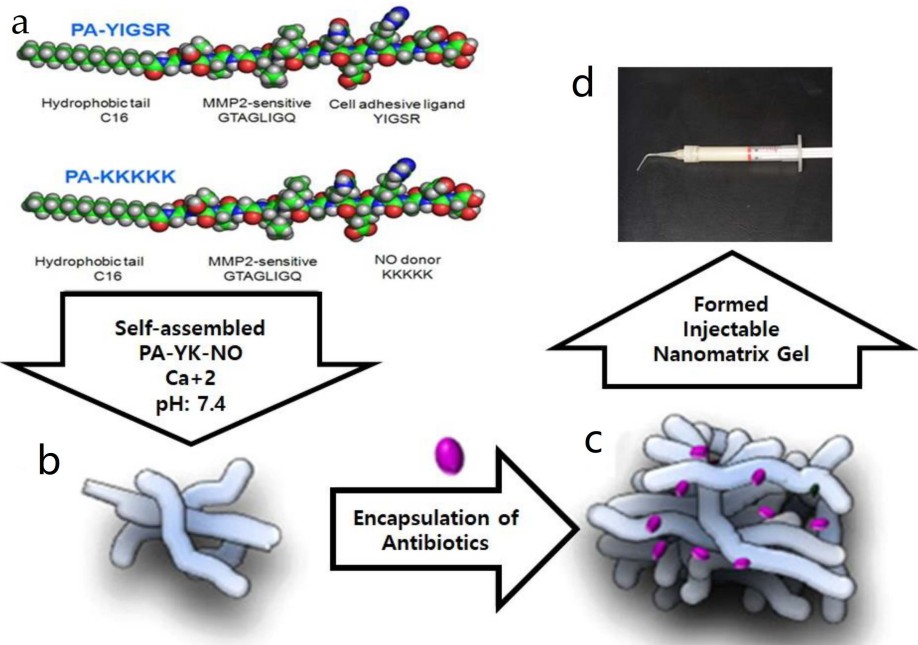

**Figure 1.** PA-YK-NO nanomatrix gel after the gelation process. (**a**) Synthesis of peptide amphiphiles (PAs) using PA-YIGSR [CH3(CH2)14CONH-GTAGLIGQ-YIGSR] and PA-KKKKK [CH3(CH2)14CONH-GTAGLIGQ-KKKKK]. (**b**) Self-assembly of PA-YK-NO (NO-releasing nanomatrix gel), the mixture of PA-YIGSR and PA-KKKKK was reacted with NO gas. (**c**) Encapsulation of doxycycline into the PA-YK-NO. (**d**) Formation of injectable PA-YK-NO.

### 2.3. In Vivo Experiment with Rat

A total of twenty-eight male Sprague Dawley rats (Semtaco, Gyeonggi-do, Korea), aged 6 to 8 weeks and weighing approximately 150 to 200 g, were used. Rats were entrusted to the 'Laboratory for Animal Experiments' in our institution. Three rats lived in one cage. Water was supplied at a rate of 500cc once every two days, and feed was supplied at a rate of 450 g twice a week.

All surgical procedures were performed under intramuscular injection of Zoletil 50 (100–150 mg/kg; Virbac Lab, Carros, France) and underwent a 5-day treatment with 0.4% β-aminopropionitrile (β-APN; Sigma-Aldrich, St Louis, MO, USA) to aviod traumatic extraction [29]. After cleaning the oral cavity with 2% chlorhexidine solution, twenty-eight right maxillary first molars were gently extracted using a sterile extraction forceps. Roots that were fractured during the extraction were excluded. Seven teeth in each group were maintained by additional experimental animals. The extracted teeth were exposed to a dry environment for 5 min at room temperature. Then, the teeth and sockets were cleaned with saline prior to replantation, or preserved in Hanks' Balanced Salt solution (HBSS, Gibco Laboratories, Grand Island, NY, USA) for 30 min or 60 min. After replantation, all rats received a single dose of 20,000 IU of penicillin G (Alvogen potassium penicillin G, Alvogen, Seoul, Korea) by intramuscular injection. A soft diet was fed for 7 days. All animals were sacrificed at 8 weeks after replantation.

Twenty-eight maxillary first molars with complete root formation were allocated into (1) Group I (root surface treatment with doxycycline-loaded NO gel) or (2) Group II (no surface treatment). Group I was divided into 2 sub-groups (*n* = 7 per group) according to storage time in HBSS: replantation after either a 30 min or 60 min storage period in HBSS. The teeth in Group I were soaked in doxycycline (Hana Pharm Co., Seoul, Korea) 1mg per 20 mL of PA-YK-NO gel for 5 min prior to replantation. Additionally, Group II was divided into 2 sub-groups (*n* = 7 per group) according to the same conditions: replantation after a 30 min or 60 min storage period in HBSS.

*2.4. Histomorphometric Analysis*

The specimens were washed, dehydrated, embedded in paraffin, and sectioned in the transverse plane from the middle third of the mesiobuccal root. The sectioned specimens were stained with hematoxylin and eosin (H&E). Additionally, immunohistochemical (IHC) staining was performed using the IHC marker CD31 (CD31 Polyclonal antibody; rabbit polyclonal antibody, bs-0468R, Bioss Inc., Woburn, MA, USA). After incubation with the primary antibodies, the sections were incubated for 30 min with a mouse monoclonal antibody against CD31. After rinsing in phosphonate buffered saline, the sections were incubated at room temperature for 30 min and then stained with 3,3′–diaminobenzidine (DAB) for 15 min. The sectioned specimens were scanned and digitalized (Panoramic 250 Flash III, 3DHISTECH, Budapest, Hungary); this was followed by fixation in 10% buffered formalin and decalcification with 10% formic acid.

Histomorphometric analysis was quantified using digitalized H&E-stained images using CaseViewer version 2.0 software (3DHistech, Ltd., Budapest, Hungary). The analysis was performed by two experienced and trained examiners who were blinded to the group allocation. The analysis included the following parameters: (1) pulp regeneration (area of pulp regeneration without pulp inflammation; area of calcified mass was also included) and (2) pulp inflammation (area of pulp inflammation). Both pulp regeneration and pulp inflammation were graded on a 4-point scale for each parameter (Table 1).

**Table 1.** Histomorphometric criteria used in this study.

| Grade | Description and Degree |
|:-----:|:----------------------:|
| 0 | Not observed pulp inflammation or regeneration |
| 1 | Less than 25% observed (<25%) |
| 2 | Less than 50% observed (<50%) |
| 3 | Less than 75% observed (<75%) |
| 4 | More than 75% observed ($\geq$75%) |

The analysis was performed by two experienced and trained examiners who were blinded to group allocation. Prior to the analysis, intra-examiner calibration was performed under the supervision of a senior investigator. For assessing the consistency of inter-examiners, intra-class coefficient (ICC) values were calculated for the aforementioned parameters for all images. The values of Cohen's kappa indicated acceptable inter-examiner reliability (0.560–0.975).

*2.5. Statistical Analysis*

Histomorphometric results were recorded in the format of mean $\pm$ standard deviation, and data were analyzed using SPSS 15.0 software (SPSS Inc., Chicago, IL, USA). Mann–Whitney U test ($p < 0.05$) was used to statistically compare the regenerative effect between groups.

## 3. Results

All specimens presented various types of pulp responses. Healing patterns are shown in Figures 2 and 3. Histological morphology of specimens was altered in both composition and density of pulp tissues after replantation. Healing of pulp tissues underwent the following patterns: (1) normal pulp with disorganization of odontoblastic layer; (2) internal reparative dentin formation with connective tissue, bone-like, and/or cementum-like tissue; and (3) pulp necrosis.

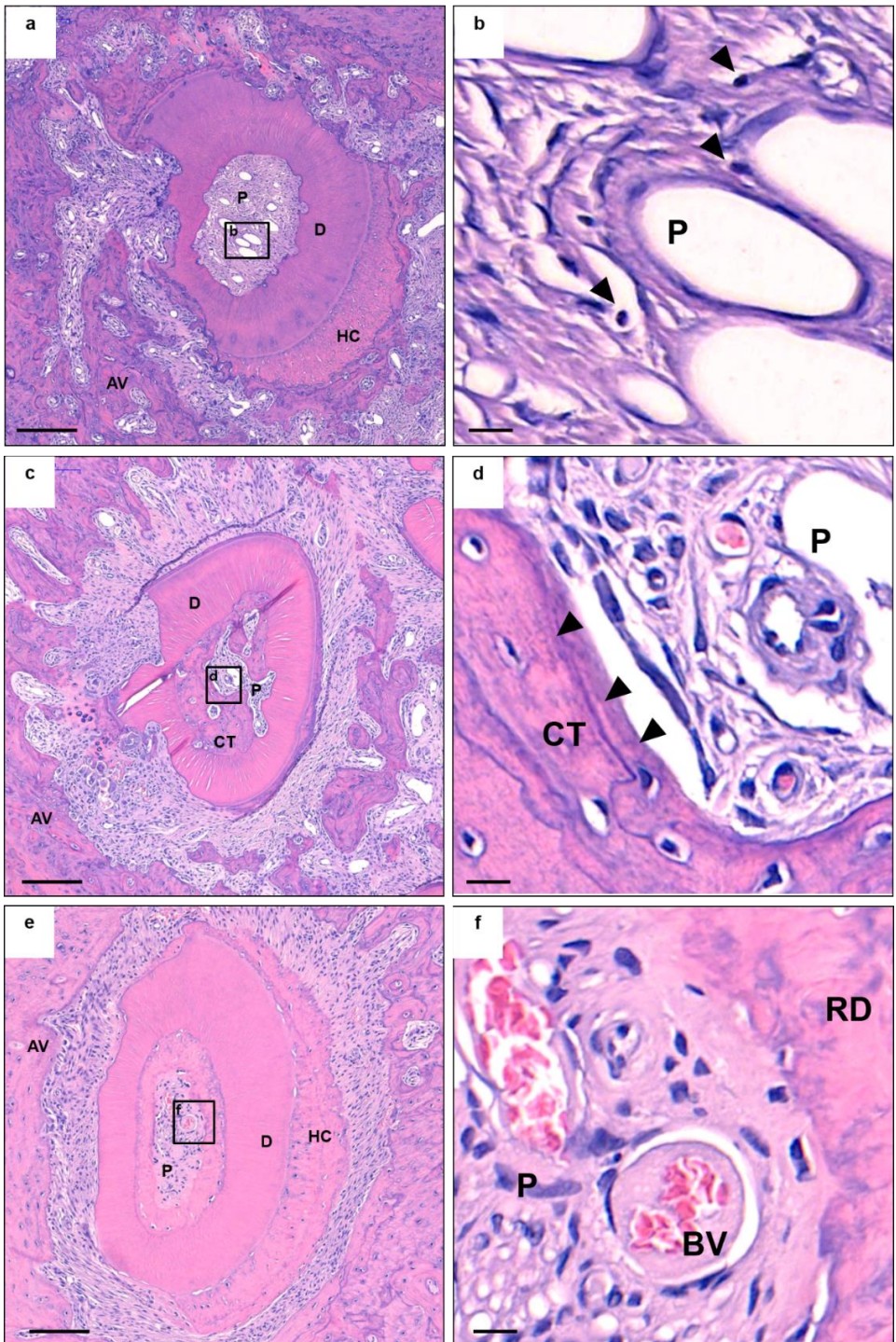

**Figure 2.** Histologic observations of specimens. (**a**,**c**,**e**) are hematoxylin and eosin (H&E) stains of the experimental group, and (**b**,**d**,**f**) are high-magnification images of the squares in (**a**,**c**,**e**) respectively. (**a**) Pulp necrosis is observed. (**b**) Lymphocytes (arrows) and fibroblasts are observed in pulp proper. (**c**) Formation of calcified tissue in pulp is observed. A severe degree of external root resorption is progressed. (**d**) Extensive amount of bone-like tissue (arrows) is formed in pulp. Presence of blood vessel proliferation is visible. (**e**) Pulp regeneration is shown in pulp proper. (**f**) Disruption of the odontoblastic layer and formation of reparative dentin layer is visible. However, pulp proper is normal. Scale bars = 200 μm for (**a**,**c**,**e**) and 10 μm for (**b**,**d**,**f**). AV, alveolar bone; D, dentin; P, pulp proper; HC, hypercementosis; CT, calcified tissue; RD, reparative dentin; BV, blood vessel.

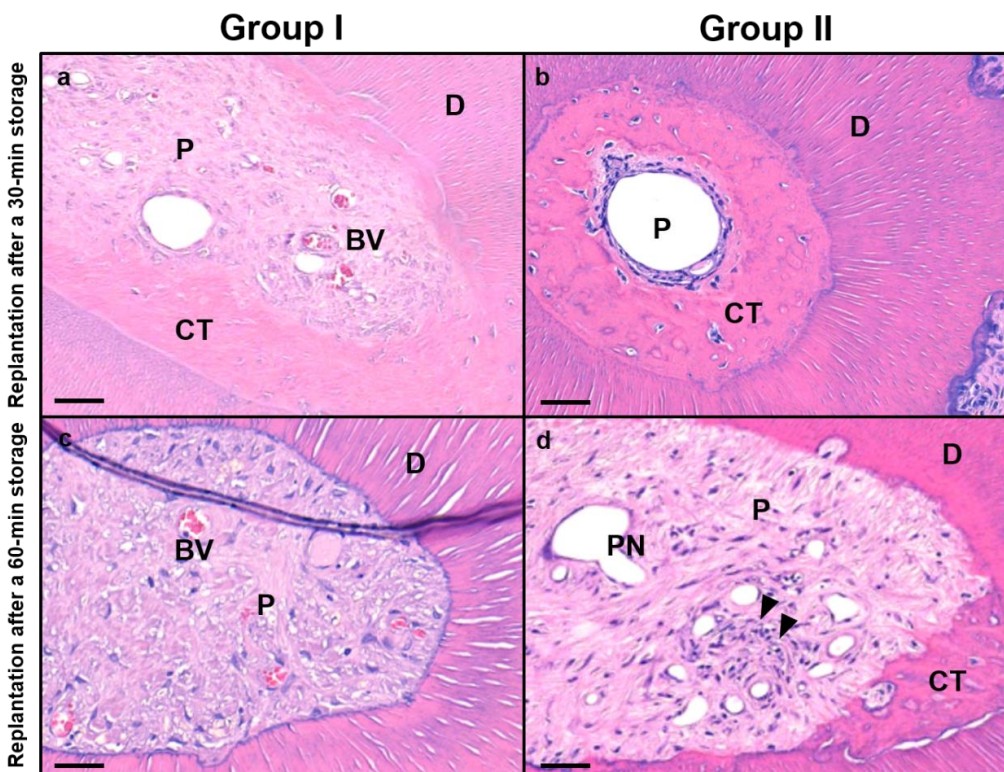

**Figure 3.** Histologic sections of teeth according to groups. (**a**,**c**) H&E stains of Group I and (**b**,**d**) those of Group II. D, dentin; P, pulp proper; CT, calcified tissue; BV, blood vessel; PN, pulp necrosis; black arrow, lymphocyte. Scale bars = 100 μm.

The intraclass correlation coefficient (ICC) values indicated acceptable inter-examiner reliability in the histomorphometric analysis (ICC = 0.560–0.975). No significant differences between Group I and II were found in pulp regeneration in replantation after 30 min storage (Table 2). However, Group I showed a significantly lower grade of pulp inflammation compared to Group II ($p < 0.05$). Figure 4a–h shows IHC morphologies of samples of two groups. Group I showed lower grade of infiltration of macrophages compared to Group II (Figure 4a,b,e,f). Similar amounts of CD31-positive cells were observed in the two groups (Figure 4c,d,g,h).

**Table 2.** Histomorphometric analysis of the replantation after 30 min storage. There was no difference in the area of pulp regeneration in Group I and II, but there was a statistically significant difference in the area of pulp inflammation in Group I and II.

|  | **Group I** | **Group II** | *p*-**Value** |
|---|---|---|---|
| Pulp regeneration | 2.43 ± 0.85 | 2.21 ± 1.37 | 0.721 |
| Pulp inflammation | 1.00 ± 1.11 | 2.21 ± 1.25 | 0.014 * |

* = statistically significant ($p < 0.05$, Mann–Whitney U test).

In groups of replantation after a 60 min storage period, a significantly higher grade of pulp regeneration occurred in Group I, and a lower grade of pulp inflammation was observed in Group I ($p < 0.05$) (Table 3). Figure 4i–p shows IHC morphologies of samples of two groups. A marked degree of vessel proliferation was observed in Group I, whereas various degrees of infiltration of macrophages were observed in Group II. In addition, the healed pulp tissue in one sample of Group I was closely similar with a normal pulp tissue (Figure 4k,l).

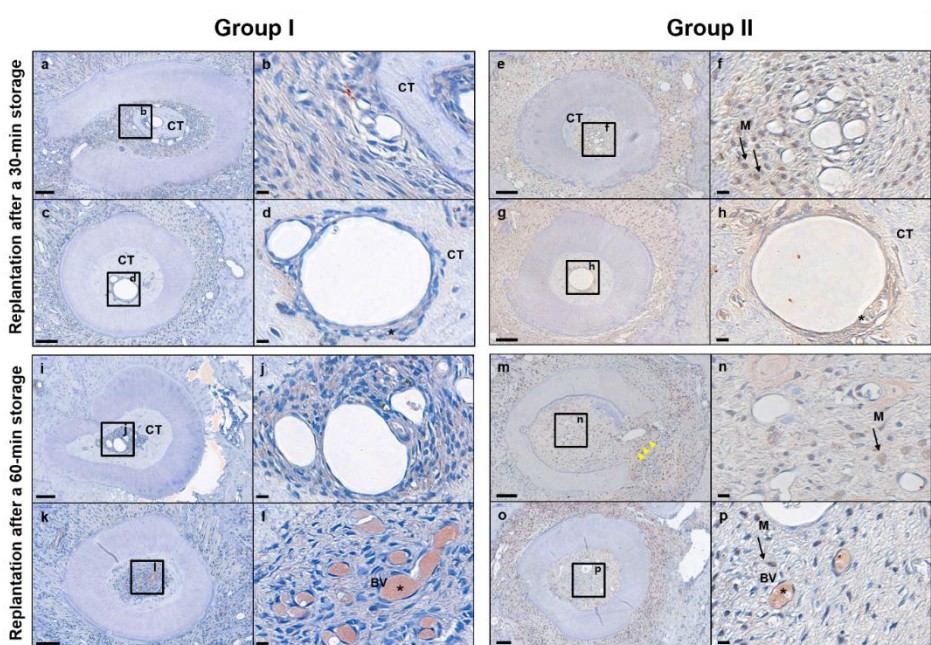

**Figure 4.** Immunohistochemical (IHC) analysis. (**a–d,i–l**) IHC stains of Group I and (**e–h,m–p**) IHC stains of Group II. (**b,d,f,h,j,l,n,p**) High-magnification images of pulp proper of (**a,c,e,g,j,k,m,o**), respectively. (**a,b**) Healing with calcified tissue is observed. Infiltration of macrophages is hardly visible compared to Group II. (**c,d**) CD31-positive cells (asterisks) are visible around the vessel. Furthermore, an extensive amount of calcified tissue is formed around vessel. (**e,f**) Broad infiltration of macrophages is visible. (**g,h**) CD31-positive cells are visible around the vessel. An extensive amount of calcified tissue is formed around the vessel. (**i,j**) Healing with calcified tissue is observed. Additionally, infiltration of macrophages is hardly visible compared to the control group. (**k,l**) Regenerated pulp tissue similar to normal pulp tissue is visible. Greater vessel proliferation is observed compared to the control group. (**m,n**) Complete destruction of root with severe degree of inflammation (yellow arrows). Infiltration of macrophages in pulp proper is also visible. (**o,p**) Vessel proliferation is observed with mild amount of infiltration of macrophages. Scale 190 bars = 100 μm for (**a,c,e,g,i,k,m,o**) and 10 μm for (**b,d,f,h,j,l,n,p**). CT, calcified tissue; M, macrophage; BV, blood vessel.

**Table 3.** Histomorphometric analysis of the replantation after 60 min storage. There were statistical differences in the areas of pulp regeneration and pulp inflammation in Groups I and II.

|  | Group I | Group II | *p*-Value |
|---|---|---|---|
| Pulp regeneration | $2.57 \pm 0.85$ | $1.57 \pm 1.02$ | 0.012 * |
| Pulp inflammation | $1.00 \pm 0.68$ | $1.57 \pm 0.51$ | 0.025 * |

* = statistically significant ($p < 0.05$, Mann–Whitney U test).

## 4. Discussion

In the present study, the effect of root surface treatment with doxycycline-loaded NO-releasing nanomatrix gel was evaluated for pulp regeneration in replantation of avulsed rat teeth. Because the purpose of this study was to evaluate regenerative potentials for pulp tissues, only avulsed teeth preserved within 1 h in storage media were included. In spite of the avulsed tooth being preserved in an ideal storage medium, the condition of the periodontal ligament cells may be viable but compromised. In addition, an avulsed tooth with an extra-oral time of longer than 1 h cannot maintain its vitality [4,30].

The duration of storage time after the avulsion may affect cellular viability. In the present study, the grade of pulp regeneration in Group I of replantation after 60 min storage was significantly increased compared to Group II. Additionally, greater vessel proliferation was observed in Group I of replantation after 60 min storage. However, the grade of pulp regeneration did not show significant differences between the two groups of replantation

after a 30 min storage period. A possible explanation may include different estimated viable cell numbers according to storage duration. Pulp cells preserved in a 30 min storage media are highly likely to maintain their viabilities compared to pulp cells in a 60 min storage media. Thus, it was shown that pulp cells preserved in a storage media for 30 min can have greater regenerative potential.

Pulp tissues of Group I healed with the following three patterns: normal pulp with disorganization of odontoblastic layer; internal reparative dentin formation with connective tissue, bone-like, and/or cementum-like tissue; and pulp necrosis. Our results are different from those of previous replantation studies of doxycycline, which reported that some replanted teeth treated with doxycycline healed with normal pulp tissue and reestablishment of intact odontoblastic layer [6,31]. This comparison may have been affected by different durations of extra-oral time rather than the effect of PA-YK-NO gel. The previous studies were designed to reproduce clinical conditions of immediate replantation. Thus, the injured pulp tissues were more likely to regenerate because all replanted teeth in the previous studies were exposed for a few seconds to minutes [7,29]. Even if the injured pulp tissue healed with hard tissue formation, it should not be regarded as a failure of pulp regeneration. Pulp canal obliteration is the compensatory response of vital pulp to a severe injury, and it is generally accepted that the incidence of pulp necrosis following pulp canal obliteration is very low [32].

Dental pulp tissues appeared to be healed with fibrous, osteoid, and cementoid tissue formation in the present study. These results corresponded with other studies, suggesting that newly formed pulp tissues after regenerative endodontic therapy are composed of periodontal-like, bone-like, and cementum-like tissues, rather than dentin [33–35]. These findings support that the periodontal ligament, bone, and cementum may grow into the pulp canal during the healing process after replantation [35]. Several studies suggested that this healing response may initiate and progress in the absence of pulp tissue in the pulp canal [36,37]. Moreover, bone-like tissues were deposited in the case of a large amount of external root resorption extending beyond the inner dentin surface (Figure 2c,d). It is possible that the endogenous cells located around the root surface were recruited into the pulp canal and differentiated to bone/cementum-like cells because direct communication into the pulp canal was easily available.

Doxycycline has been used in attempts to promote healing of avulsed teeth after replantation, and some authors reported a positive effect of doxycycline on pulp revascularization after replantation [5–7]. They suggested that this effect is associated with the antimicrobial activity of doxycycline because doxycycline could decrease the number of micro-organisms on the root surface during extra-oral time. However, there is controversy about the effect of doxycycline on pulp regeneration after replantation. Some studies reported that doxycycline had no effect on micro-organisms in necrotic pulp. Doxycycline only had an effect on inflammatory inhibition, such as reducing the occurrence of inflammatory resorption or ankylosis [38]. The direct effect of doxycycline on angiogenic capacity is not fully identified in the pulp tissue of an avulsed tooth [24]. Therefore, angiogenic regenerative potential could be achieved using NO-releasing nanomatrix gel.

NO is known for its bio-active functions such as the promotion of angiogenesis and subsequent development of mature blood vessels, along with the release of vascular endothelial growth factor (VEGF) [39]. Furthermore, NO inhibits apoptosis of vascular endothelial cells induced by proinflammatory cytokines and proatherosclerotic factors, including angiotensin II [23]. Apoptosis of endothelial cells may contribute to the development of inflammatory processes. Thus, NO can inhibit inflammatory processes. Our results indicated that NO-releasing nanomatrix gel decreases the degree of pulp inflammation.

The study includes several limitations. The sample size was rather small because the study was designed as a pilot study; future studies will be verified further using an increased sample size. The regenerative potential of NO-releasing nanomatrix gel is reported to have therapeutic functions in bacterial infections, would healing, and cardiovascular diseases [21,22,39]. However, the interaction between the NO-releasing nanomatrix gel and

doxycycline is not reported yet; therefore, future studies will be performed to characterize the conditions in vitro, such as optimal concentrations of NO and doxycycline. It is difficult to predict the effect of doxycycline and NO on pulp regeneration after in vivo replantation and follow-up. Thus, the present study referenced the previous regimen of doxycycline with saline and modified the regimen using NO-releasing nanomatrix gel as a substitute for saline [4]. Further investigations are necessary to identify whether doxycycline and NO have synergistic/antagonistic interactions or independent interactions.

In conclusion, the present study demonstrates that doxycycline-loaded NO-releasing nanomatrix gel is associated with the promotion of pulp regeneration following replantation of avulsed rat teeth. Our findings suggest that NO gel may provide an environment that enhances the pulp cell.

**Author Contributions:** Conceptualization, K.-H.Y.; methodology, K.-H.Y.; software, O.-H.N.; validation, M.-J.K., Y.-K.C.; formal analysis, K.C., K.L.; investigation, S.-C.C.; resources, S.-C.C.; data curation, H.-S.L.; writing—original draft preparation, K.-H.Y.; writing—review and editing, H.-S.L., K.C., S.-C.C.; visualization, O.-H.N.; supervision, S.-C.C.; project administration, K.C.; funding acquisition, S.-C.C. and K.C. All authors have read and agreed to the published version of the manuscript.

**Funding:** This research was supported by Basic Science Research Program through the National Research Foundation of Korea (NRF) funded by the Ministry of Education (NRF-2016R1D1A1B03931063) and National Institutes of Health/National Institute of Dental and Craniofacial Research, K08 (KDE027401A).

**Institutional Review Board Statement:** The study was reviewed and approved by the Ethics in Institutional Animal Care and Use Committee of Kyung Hee Medical Center, Kyung Hee University, Seoul, Korea (KHMC-IACUC-16-022).

**Informed Consent Statement:** Not applicable.

**Data Availability Statement:** The data presented in this study are openly available.

**Acknowledgments:** Thanks to everyone that helped in this study.

**Conflicts of Interest:** The authors declare no conflict of interest.

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
