# Peer review of "Doxycycline-Loaded Nitric Oxide-Releasing Nanomatrix Gel in Replanted Rat Molar on Pulp Regeneration"

_applsci, doi:10.3390/app11136041_

Round 1

Reviewer 1 Report

Nitric oxide (NO) is very important compound of our organisms. NO regulates inflammatory processes and promotes angiogenesis. In the reviewed paper, Authors presented that No-releasing nanomatrix gel improved tooth pulp regeneration. Results are important for dentistry and tooth replantation. I would like suggest some corrections:

  1. English must be corrected by native, because are frequent linguistic mistakes.
  2. Materials and Methods should be divided into subsections.
  3. Please write, which company produced NO-releasing nanomatrix gel.
  4. How and what were the rats fed?
  5. In Tables 3 and 4 should be added units. The tables need to be self-contained, and now it is not known what these results mean, no units or way of explanations. Are they volume, thickness, amount, uptake, or grade of inflammation?
  6. On what guidelines was the grade of inflammation developed?

Author Response

Nitric oxide (NO) is very important compound of our organisms. NO regulates inflammatory processes and promotes angiogenesis. In the reviewed paper, Authors presented that No-releasing nanomatrix gel improved tooth pulp regeneration. Results are important for dentistry and tooth replantation. I would like suggest some corrections:

  1. English must be corrected by native, because are frequent linguistic mistakes.
  • We did the language revision. Thank you.

  1. Materials and Methods should be divided into subsections.
  • We divided the Materials and Methods into subsections.

  1. Please write, which company produced NO-releasing nanomatrix gel.
  • We synthesized the NO-releasing nanomatrix gel as the previous study (reference 26).

  1. How and what were the rats fed?
  • We added the information about feeding rats.

  1. In Tables 3 and 4 should be added units. The tables need to be self-contained, and now it is not known what these results mean, no units or way of explanations. Are they volume, thickness, amount, uptake, or grade of inflammation?
  • We are very sorry for the lack of explanation. I added a description to the table. These numbers are the average and statistical analysis results of the values measured using the criteria presented in Table 2.

  1. On what guidelines was the grade of inflammation developed?
  • The method of measuring pulp regeneration and pulp inflammation by area was devised by the authors of this study, and good results were obtained by performing inter-examiner reliability.

Reviewer 2 Report

This work tests the hypothesis that a nitric oxide-releasing nanomatrix gel enhances pulp regeneration upon replantation. The study is well designed, but the sample size is rather small and it is unclear why only one tooth has been replanted in each rat. This limitation does not disqualify the study, but a careful revision is necessary to make the paper clear and informative. 

1. The paragraph located between lines 256 and 265 fits in the "Introduction", in the region of lines 47-51. It provides background information that serves to explain the strategy of the present study. 

2. The "Materials and Methods" section should be extended to explain the statistical analysis in detail. For example, it is unclear which type of intraclass correlation coefficient (ICC) is reported in results (line 165). For the classification of ICCs, see, e.g. [Shrout PE, Fleiss JL. Intraclass correlations: uses in assessing rater reliability. Psychol Bull. 1979;86(2):420-8.] and [Koo TK, Li MY. A Guideline of Selecting and Reporting Intraclass Correlation Coefficients for Reliability Research. J Chiropr Med. 2016;15(2):155-63.]. Also, explain the details of the reliability evaluation: how many repeated measurements were taken by how many observers? Did they involve the entire sample?  

3. The illustrations need more precise and complete annotations. The regions shown as high magnification images in Figure 2b,d are marked imprecisely in Figure 2a,c, respectively. They are considerably smaller than indicated by the frames placed in Figure 2a,c.  Please remedy and add various markers to Figure 2b,d to illustrate the statements from the figure legend (e.g. add an arrow to show the dentin layer of panel f). Similar problems are present in Figure 3. Make sure that all the regions of interest are precisely marked on the low-magnification pictures and add explanatory markers on the high-magnification images. 

4. Please add one more figure of HE staining with the structure similar to Figure 3, showing representative sections of teeth from each group. Another option would be to replace Figure 2 with a more complex, 16-panel figure of H&E sections.  

5. Although the language is comprehensible, it needs refinement, especially in the "Discussion" section.  This section is essential for demonstrating the relevance of the study in connection with the efforts of other laboratories. 

Minor remarks: 

Line 17: In the Abstract, please state that data are reported as mean±standard deviation, and specify the P-values obtained in the statistical hypothesis test. Also, the degree of pulp inflammation is reported twice (on lines 17 and 19). Unless it is explained that the two results differ in the time of storage in Hanks’ Balanced Salt solution, the reader might suspect that the reporting is repeated and inaccurate. 

Line 21:  I would add one sentence explaining the limits of the study. Otherwise, the last sentence of the Abstract remains vague. 

Line 50: "18].Moreover, No" => "18]. Moreover, NO"

Line 87: Please add a legend after the title of Figure 1 and explain what is represented in the 4 panels (A-D) and in the image on the right.   
Line 91: Please define each acronym before its first use: instead of "HBSS", please write "Hanks’ Balanced Salt solution (HBSS)".

Line 98: Table 1 is redundant: it merely repeats the information given in the paragraph that precedes it. I would remove it. 

Line 109: maintaiend => maintained
Also, please revise this sentence to clarify what happened in the case of broken roots observed after extraction. 

Line 152: The title of Figure 2 is imprecise. It could be revised as "Typical histological sections of extracted root specimens of group ...".

Line 228-230: Please revise this sentence. Terms such as "periodontal-like, bone-like, and cementum-like" are non-specific. 

Line 240: Instead of "some authors reported", please provide references.

Line 275: Please mention the small sample size in the paragraph dedicated to the limitations of the study.  

Author Response

Reviewer 2

This work tests the hypothesis that a nitric oxide-releasing nanomatrix gel enhances pulp regeneration upon replantation. The study is well designed, but the sample size is rather small and it is unclear why only one tooth has been replanted in each rat. This limitation does not disqualify the study, but a careful revision is necessary to make the paper clear and informative. 

  1. The paragraph located between lines 256 and 265 fits in the "Introduction", in the region of lines 47-51. It provides background information that serves to explain the strategy of the present study. 

=> We relocated the paragraph. Thank you.

  1. The "Materials and Methods" section should be extended to explain the statistical analysis in detail. For example, it is unclear which type of intraclass correlation coefficient (ICC) is reported in results (line 165). For the classification of ICCs, see, e.g. [Shrout PE, Fleiss JL. Intraclass correlations: uses in assessing rater reliability. Psychol Bull. 1979;86(2):420-8.] and [Koo TK, Li MY. A Guideline of Selecting and Reporting Intraclass Correlation Coefficients for Reliability Research. J Chiropr Med. 2016;15(2):155-63.]. Also, explain the details of the reliability evaluation: how many repeated measurements were taken by how many observers? Did they involve the entire sample?  

=> The analysis was performed by two experienced and trained examiners who were blinded to the group allocation. In histomorphometric analysis, the intraclass correlation coefficient (ICC) values indicated acceptable inter-examiner reliability in histomorphometric analysis (ICC=0.560-0.975).

  1. The illustrations need more precise and complete annotations. The regions shown as high magnification images in Figure 2b,d are marked imprecisely in Figure 2a,c, respectively. They are considerably smaller than indicated by the frames placed in Figure 2a,c.  Please remedy and add various markers to Figure 2b,d to illustrate the statements from the figure legend (e.g. add an arrow to show the dentin layer of panel f). Similar problems are present in Figure 3. Make sure that all the regions of interest are precisely marked on the low-magnification pictures and add explanatory markers on the high-magnification images. 

=> Thank you. We corrected the Figure 2.

  1. Please add one more figure of HE staining with the structure similar to Figure 3, showing representative sections of teeth from each group. Another option would be to replace Figure 2 with a more complex, 16-panel figure of H&E sections.  

=> Thank you. We added one more figure of HE staining as Figure 3.

  1. Although the language is comprehensible, it needs refinement, especially in the "Discussion" section.  This section is essential for demonstrating the relevance of the study in connection with the efforts of other laboratories. 

=> We did the language revision again. Thank you.

Minor remarks: 

Line 17: In the Abstract, please state that data are reported as mean±standard deviation, and specify the P-values obtained in the statistical hypothesis test. Also, the degree of pulp inflammation is reported twice (on lines 17 and 19). Unless it is explained that the two results differ in the time of storage in Hanks’ Balanced Salt solution, the reader might suspect that the reporting is repeated and inaccurate. 

  • We added mean ± standard deviation and p-value. However, there is pulp regeneration between the two pulp inflammation. Removing the pulp inflammation can cause confusion in the content. Thank you for your understanding.

Line 21:  I would add one sentence explaining the limits of the study. Otherwise, the last sentence of the Abstract remains vague. 

  • We added the limit of the study.

Line 50: "18].Moreover, No" => "18]. Moreover, NO"

  • Thank you.

Line 87: Please add a legend after the title of Figure 1 and explain what is represented in the 4 panels (A-D) and in the image on the right.

  • We corrected it.

Line 91: Please define each acronym before its first use: instead of "HBSS", please write "Hanks’ Balanced Salt solution (HBSS)".

  • We corrected it.

Line 98: Table 1 is redundant: it merely repeats the information given in the paragraph that precedes it. I would remove it. 

  • We removed it.

Line 109: maintaiend => maintained
Also, please revise this sentence to clarify what happened in the case of broken roots observed after extraction. 

  • We revised it.

Line 152: The title of Figure 2 is imprecise. It could be revised as "Typical histological sections of extracted root specimens of group ...".

  • We revised it as you guided.

Line 228-230: Please revise this sentence. Terms such as "periodontal-like, bone-like, and cementum-like" are non-specific. 

  • We revised it.

Line 240: Instead of "some authors reported", please provide references.

  • We corrected.

Line 275: Please mention the small sample size in the paragraph dedicated to the limitations of the study.  

  • We revised it.

Reviewer 3 Report

The manuscript describes the bone regeneration potential of a doxycycline loaded NO releasing hydrogel in an avulsed tooth setting. The teeth treated with the hydrogel promoted better bone-like formation than the control group after 8 weeks of replantation. Here are some suggestions which could help improve the manuscript:

Add in the term “doxycycline loaded” to the title.

Line 14: Not sure what “use or root surface treatment” means. Please correct “or” to “of” if it is a typographical error.

Line 17-19: lower grade of pulp inflammation is mentioned twice. Please correct it.

Line 40-43: Please break down the sentence into two smaller sentences for ease of reading.

In certain occasions, nitric acid-NO- has been abbreviated wrongly as No. Please revise that.

Line 50: Remove “activity in”

Line 55: Replace “As” with “they”.

Line 55-56: Add reference to the sentence “In addition, they can form a viscoelastic gel by themselves in certain circumstances.”

Line 62-63: The sentence “A doxycycline…developed” is not required.

Line 70: the term “NO releasing” is repeated twice.

Line 78: Why 9:1 ratio. Provide explanation in the discussion section.

In figure 1A please replace RGDS with YIGSR and “self-assembled PA-RK-NO..” with “self-assembled PA-YK-NO..”.

Please add some text about the method the antibiotic was encapsulated in the gel.

Please specify what section of the rat’s craniofacial region was harvested.

In figure 2, specify if the images represent the 30 minute or 60 minute experimental groups. If no difference was observed, please mention it in the text. In Figure 2b, indicate the blood vessels and in figure 2f, indicate the dentin layer. Add in histology sections of control groups too (atleast as supplementary information)

Line 64: Please expand ICC, PDL.

In fig 3, please indicate the macrophages, CD31+ cells, new bone formation, and blood vessels with arrows or similar markings.

Line 220- not sure if in a clinical setting there would be an immediate replantation.

Line 246: Do not start a sentence with “And”.

Line 248-249: add in reference for doxycycline’s angiogenic potential.

In the paragraph in lines 266- 274, compare the 30 minute and 60 minute subgroups within each experimental and control groups. And mention that the cells in the 60 minute experimental group had greater viability because of the nanomatrix gel.

Discuss about the release profile of doxycycline and NO. In ref 25, there is a release profile of NO from films made of the same peptide assembly. Authors might want to elaborate on that or talk about how the release would be different from a gel system. Authors express their concern about predicting the drug release in a delayed replantation setting, but there might be some advantage to performing a regular in vitro release study, to atleast understand any possible interaction between the co-release of NO and doxycycline.

Discuss what would be the effect if the concentration of the drug or NO were altered (increased/ decreased).

Not sure why the authors are using the term “doxycycline mediated”. Was doxycycline somehow responsible for NO release? If so, please elaborate. If not, edit it to “doxycycline loaded”.

Some language errors in the Discussion section need to be addressed/ proofread.

Author Response

Reviewer 3

The manuscript describes the bone regeneration potential of a doxycycline loaded NO releasing hydrogel in an avulsed tooth setting. The teeth treated with the hydrogel promoted better bone-like formation than the control group after 8 weeks of replantation. Here are some suggestions which could help improve the manuscript:

Add in the term “doxycycline loaded” to the title.

  • We corrected as you suggested.

Line 14: Not sure what “use or root surface treatment” means. Please correct “or” to “of” if it is a typographical error.

  • We corrected as you suggested.

Line 17-19: lower grade of pulp inflammation is mentioned twice. Please correct it.

  • There is pulp regeneration between the two pulp inflammation. Removing the pulp inflammation can cause confusion in the content.
  • We hope you understand.

Line 40-43: Please break down the sentence into two smaller sentences for ease of reading.

  • We corrected as you suggested.

In certain occasions, nitric acid-NO- has been abbreviated wrongly as No. Please revise that.

  • We corrected as you suggested.

Line 50: Remove “activity in”

  • We corrected as you suggested.

Line 55: Replace “As” with “they”.

  • We corrected as you suggested.

Line 55-56: Add reference to the sentence “In addition, they can form a viscoelastic gel by themselves in certain circumstances.”

  • We corrected as you suggested.

Line 62-63: The sentence “A doxycycline…developed” is not required.

  • We corrected as you suggested.

Line 70: the term “NO releasing” is repeated twice.

  • We corrected as you suggested.

Line 78: Why 9:1 ratio. Provide explanation in the discussion section.

  • We followed previous ratio on reference #22(Kushwaha, M.; Anderson, J.M.; Bosworth, C.A.; Andukuri, A.; Minor, W.P.; Lancaster, J.R., Jr.; Anderson, P.G.; Brott, B.C.; Jun, 370 H.W. A nitric oxide releasing, self assembled peptide amphiphile matrix that mimics native endothelium for coating 371 implantable cardiovascular devices. Biomaterials 2010, 31, 1502-1508) : page 1504; Optimization of PA-YIGSR and PA-KKKK.

In figure 1A please replace RGDS with YIGSR and “self-assembled PA-RK-NO..” with “self-assembled PA-YK-NO..”.

  • We corrected it.

Please add some text about the method the antibiotic was encapsulated in the gel.

  • 1mg of doxycycline powder dissolved in 10mL of PA-YK-NO gel to adjust the proper viscosity.

Please specify what section of the rat’s craniofacial region was harvested.

  • The specimens were washed, dehydrated, embedded in paraffin, and sectioned in the transverse plane from the middle third of the mesiobuccal root of 1st maxillary molar.

In figure 2, specify if the images represent the 30 minute or 60 minute experimental groups. If no difference was observed, please mention it in the text. In Figure 2b, indicate the blood vessels and in figure 2f, indicate the dentin layer. Add in histology sections of control groups too (atleast as supplementary information)

  • The Figure 2 showed a typical histologic observation. We removed the ‘Group 1’ from the figure legend.

Line 64: Please expand ICC, PDL.

  • We corrected as you suggested.

In fig 3, please indicate the macrophages, CD31+ cells, new bone formation, and blood vessels with arrows or similar markings.

  • We corrected it.

Line 220- not sure if in a clinical setting there would be an immediate replantation.

  • In recent years, knowledge about replantation has become public, so parents or health teacher may replant the avulsed teeth immediately.

Line 246: Do not start a sentence with “And”.

  • We corrected as you suggested.

Line 248-249: add in reference for doxycycline’s angiogenic potential.

  • We corrected as you suggested.

In the paragraph in lines 266- 274, compare the 30 minute and 60 minute subgroups within each experimental and control groups. And mention that the cells in the 60 minute experimental group had greater viability because of the nanomatrix gel.

  • Within the limit of this study, the grade of pulp regeneration did not show significant differences between two groups of replantation after a 30-min. However the grade of pulp regeneration in Group I of replantation after 60-min storage was significantly increased compared to Group II. Also, greater vessel proliferation was observed in Group I of replantation after 60-min storage. This result shows that there is no significant difference in the viabilities of pulp cells in a 30-min storage time, but it is thought that there may be a difference in the regeneration capacity of pulp cells due to ischemic conditions at 60 min storage, and NO-gel treatment improves the regenerative capacity.

Discuss about the release profile of doxycycline and NO. In ref 25, there is a release profile of NO from films made of the same peptide assembly. Authors might want to elaborate on that or talk about how the release would be different from a gel system. Authors express their concern about predicting the drug release in a delayed replantation setting, but there might be some advantage to performing a regular in vitro release study, to atleast understand any possible interaction between the co-release of NO and doxycycline.

Discuss what would be the effect if the concentration of the drug or NO were altered (increased/ decreased).

  • We mentioned about the issue in the Discussion and corrected it.
  • The present study has several limitations. At first, the sample size was rather small. This is the common limitation of animal study due to limitation of the Ethics in Institutional Animal Care. Secondly the regenerative potential of NO-releasing nanomatrix gel was identified, but it is not clearly evaluated whether the potential is due to NO-releasing nanomatrix gel itself or the interaction between the gel and doxycycline. At last, there remain concerns about appropriate concentration of NO in doxycycline. It was difficult to predict the appropriate concentration of doxycycline and NO for pulp regeneration after delayed replantation. So, we followed the regimen of previous doxycycline with saline and evaluated NO as a substitute for saline[4]. Further investigations are necessary to titrate the concentration of the treatment drugs and identify whether doxycycline and NO have synergistic/antagonistic interactions or independent interactions.

Not sure why the authors are using the term “doxycycline mediated”. Was doxycycline somehow responsible for NO release? If so, please elaborate. If not, edit it to “doxycycline loaded”.

  • Thank you for your suggestion. We replaced ‘mediated’ to ‘loaded’.

Some language errors in the Discussion section need to be addressed/ proofread.

  • We did the language revision again.

Round 2

Reviewer 1 Report

Authors corrected article partially. My suggestions:

  1. In Tables 2 and 3 should be added units. Authors described that studied was area, but in mm2, um2?
  2. Please add also min and max values in tables.
  3. Authors studied connection of Doxycycline + PA-YK-NO. Interesting would be effect of PA-YK-NO without doxycycline. Please add these results to the article.

Author Response

Reviewer 1

Authors corrected article partially. My suggestions:

  1. In Tables 2 and 3 should be added units. Authors described that studied was area, but in mm2, um2?

I am so sorry for this confusion. The numbers in Tables 2 and 3 are the averaged grades (nominal variables), which was measured based on the classification in Table 1. In other words, Tables 2 and 3 have no units because they were made by nominal variables based on area.

  1. Please add also min and max values in tables.
  • As described above, since it was a nominal variable, the maximum value was 4 and the minimum value was 0.

  1. Authors studied connection of Doxycycline + PA-YK-NO. Interesting would be effect of PA-YK-NO without doxycycline. Please add these results to the article.
  • Thank you for your kind suggestion. Our current study did not compare the effectiveness with or without doxycycline. According to the previous study of our research group, doxycycline improved the periodontal healing of replanted teeth stored in HBSS. [Nam OH, et al. Evaluation of the periodontal and pulpal healing of replanted rat molars with doxycycline root conditioning. J Periodontal Implant Sci. 2019;49:148-157]. The solo effect of the PA-YK-NO could be considered in our future studies.

Reviewer 2 Report

The authors addressed most of my concerns appropriately. The language has been improved and the inaccurate markings from Fig. 2 have been corrected. One major concern of mine, however, is still standing. My referee report on the original manuscript contains the following comment: 

"The "Materials and Methods" section should be extended to explain the statistical analysis in detail. For example, it is unclear which type of intraclass correlation coefficient (ICC) is reported in Results (line 165). For the classification of ICCs, see, e.g. [Shrout PE, Fleiss JL. Intraclass correlations: uses in assessing rater reliability. Psychol Bull. 1979;86(2):420-8.] and [Koo TK, Li MY. A Guideline of Selecting and Reporting Intraclass Correlation Coefficients for Reliability Research. J Chiropr Med. 2016;15(2):155-63.]. Also, explain the details of the reliability evaluation: how many repeated measurements were taken by how many observers? Did they involve the entire sample?" 

The corresponding answer of the authors is the following:  
"=> The analysis was performed by two experienced and trained examiners who were blinded to the group allocation. In histomorphometric analysis, the intraclass correlation coefficient (ICC) values indicated acceptable inter-examiner reliability in histomorphometric analysis (ICC=0.560-0.975)." 

In my opinion, this issue has not been resolved. For the sake of reproducibility, it is necessary to remedy this problem. 

Author Response

Reviewer 2

The authors addressed most of my concerns appropriately. The language has been improved and the inaccurate markings from Fig. 2 have been corrected. One major concern of mine, however, is still standing. My referee report on the original manuscript contains the following comment: 

"The "Materials and Methods" section should be extended to explain the statistical analysis in detail. For example, it is unclear which type of intraclass correlation coefficient (ICC) is reported in Results (line 165). For the classification of ICCs, see, e.g. [Shrout PE, Fleiss JL. Intraclass correlations: uses in assessing rater reliability. Psychol Bull. 1979;86(2):420-8.] and [Koo TK, Li MY. A Guideline of Selecting and Reporting Intraclass Correlation Coefficients for Reliability Research. J Chiropr Med. 2016;15(2):155-63.]. Also, explain the details of the reliability evaluation: how many repeated measurements were taken by how many observers? Did they involve the entire sample?" 

The corresponding answer of the authors is the following:  
"=> The analysis was performed by two experienced and trained examiners who were blinded to the group allocation. In histomorphometric analysis, the intraclass correlation coefficient (ICC) values indicated acceptable inter-examiner reliability in histomorphometric analysis (ICC=0.560-0.975)." 

In my opinion, this issue has not been resolved. For the sake of reproducibility, it is necessary to remedy this problem. 

  • Thanks for your kind thought regarding selection of the ICC for the study. Based on the suggested article above (J Chiropr Med. 2016;15(2):155-63), we would like to clarify the ICC from our study; it is defined as a 2-way fixed effects model, between 2 raters (mean of k raters), and measuring consistency (definition). Therefore, the ICC (=0.560-0.975) shows as little broader range than other type of ICC design, however it indicated “moderate to good” reliability.
  • In addition, more details were added to the paper. “The analysis was performed by two experienced and trained examiners who were blinded to the group allocation. Prior to the analysis, intra-examiner calibration was performed under the supervision of a senior investigator. For assessing the consistency of inter-examiners, intra-class coefficient (ICC) values were calculated in aforementioned parameters in the all images. The values of Cohen’s kappa indicated acceptable inter-examiner reliability (0.560-0.975).”

Round 3

Reviewer 1 Report

Authors corrected manuscript and explained my doubts. Recently, I recommend article for publication.

Reviewer 2 Report

The authors added further information regarding the statistical analysis. I have no further comments. 

This manuscript is a resubmission of an earlier submission. The following is a list of the peer review reports and author responses from that submission.

Round 1

Reviewer 1 Report

Nitric oxide (NO) is very important molecule of human body, which has impact on among others blood vessels, stimulation of some hormones and inflammatory processes. Authors presented that nanomatrix gel releasing NO has impact on pulp regeneration in rats. They observed that NO showed significant higher grade of pulp regeneration and significant lower grade of pulp inflammation. This observation is important for both teeth treatment, and can be used probably also in treatment of wounds of skin and mucous membranes.

In Tables 3 and 4 should be added units and description, that tables should be self-sufficient. Please add also min and max values. Authors must also correct language and punctuation. Authors studied connection of Doxycycline + PA-YK-NO. Interesting would be effect of PA-YK-NO without doxycycline. Please add these results to the article.

Reviewer 2 Report

I found the study to be straightforward and clear. I think the idea to have a nanomatrix releasing NO in a controlled manner is one of the key developments in nano-bio-technology field. 

However, the only conclusion that the study can give to the reader is that there is an improvement upon application of the combined gel and NO. The study does not quantify the release kinetics of NO, nor the effective concentration of NO, and the experimental design also doesn't have other settings like PA-YIGSR only, for comparison. The study can be better if the change in the IHC analysis result can be compared with the amount of NO loaded in the gel.

I saw that the authors already realized the issues that I've pointed out when they wrote the paragraph starting with line 274. I agree that those need further study.

Because the conclusion is very dependent on IHC result, I strongly recommend having supplementary information available online that includes the stained sections of all the samples (similar to figure 3).  

In some sentences NO is not written in capital letters, please check them.

Reviewer 3 Report

The idea of the study is very interesting and the research approach was appropriate. However, I have two scientific concerns and a few minor typos.

Typos: 

  1. line 50 - "NO" instead of "No"
  2. line 104 - "avoid"
  3. line 164 - "in histomorphometric analysis" should be removed
  4. line 209-210 - "the PDL cells" sounds much better

Questions:

  1. line 114 - what is the protocol for dehydration/embedding since we have a mixture of hard/soft dental tissues
  2. line 122-124 - what is the protocol of decalcification (duration, conditions)? since at line 129 you mention analysing the calcified mass
  3. line 266 - I would go with "increased" only, not necessarily "significant"
  4. have you observed any color changes due to doxycicline?

Looking forward to receiving a revised version of the manuscript.

Reviewer 4 Report

Dear Author,

there is interest in reviewing the paper

It is an interesting topic and a well-structured paper,, I only suggest to implement the discussions by talking about other proposed methods for pulp regeneration, such as laser, as reported by some authors, such as

Staffoli S, Romeo U, Amorim RNS, Migliau G, Palaia G, Resende L, Polimeni A. The effects of low level laser irradiation on proliferation of human dental pulp: a narrative review. Clin Ter.2017 Sep-Oct;168(5):e320-e326 

or 

Borzabadi-Farahani A.J. Effect of low-level laser irradiation on proliferation of human dental mesenchymal stem cells; a systemic review. Photochem Photobiol B. 2016 Sep;162:577-582. doi: 10.1016/j.jphotobiol.2016.07.022. Epub 2016 Jul 25.

that should be cited.